# Chromosome and Genome Diversity in the Genus *Trifolium* (Fabaceae)

**DOI:** 10.3390/plants10112518

**Published:** 2021-11-19

**Authors:** Eliška Lukjanová, Jana Řepková

**Affiliations:** Department of Experimental Biology, Faculty of Sciences, Masaryk University, 611 37 Brno, Czech Republic; eliskalukjanova@gmail.com

**Keywords:** clover, interspecific hybridization, genome size, polyploidy, synteny, chromosomal markers, cytogenetics

## Abstract

*Trifolium* L. is an economically important genus that is characterized by variable karyotypes relating to its ploidy level and basic chromosome numbers. The advent of genomic resources combined with molecular cytogenetics provides an opportunity to develop our understanding of plant genomes in general. Here, we summarize the current state of knowledge on *Trifolium* genomes and chromosomes and review methodologies using molecular markers that have contributed to *Trifolium* research. We discuss possible future applications of cytogenetic methods in research on the *Trifolium* genome and chromosomes.

## 1. Introduction

The Fabaceae family (Leguminosae, legume or bean family) is the third-largest flowering plant family, after the Asteraceae and Orchidaceae families [1,2]. It is agronomically important, as it can form a symbiotic association with nitrogen-fixing bacteria. Several species from this family serve as genetic model organisms (e.g., *Medicago truncatula* Gaertn., *Pisum sativum* L., and *Lotus japonicus* L.). With more than 250 species, the clover genus, *Trifolium*, is one of the largest genera in this family [1,2,3]. This herbaceous genus acquired its name in reference to the characteristic form of the leaf, usually consisting of three leaflets (trifoliolate), and includes both annual and perennial species occurring natively across a large range of biotopes from meadows and open woodlands to semi-deserts and mountain ridges in temperate and, to a lesser extent, subtropical regions. The genus’s origin has been estimated to have occurred in the Early Miocene, 16–23 million years ago, and its center of origin was first assumed to be in California with its subsequent spread into Asia and hence to Europe and Africa [4]. Later, a new hypothesis was proposed of clovers originating in the Mediterranean region due to their species diversity, including the diversity in their chromosome numbers, and because the greatest occurrence of their endemic species is found in this area, with a secondary center of distribution in North America and East Africa [3,5,6,7]. By contrast, native clovers are absent from Australia and Southeast Asia.

Attempts have been made to divide this genus into natural groups. In the 19th century, Bossier [8] divided the genus into seven sections. A century later, eight subgenera were recognized and revised [1,9]. Better insight into phylogeny and the origin of the genus was facilitated by molecular analyses. These showed that *Trifolium* is a member of a large clade of legumes that lack one copy of the chloroplast inverted repeat [10,11], and a further molecular phylogenetic analysis of the internal transcribed spacer (ITS) and chloroplast genes provided evidence that most of these proposed sections are not monophyletic [12,13]. The most recent subgeneric classification, based on phylogenetic analyses of 218 species’ ribosomal ITS and chloroplast trnL intron sequences, was proposed by Ellison et al. [3], who divided the genus into two subgenera, *Chronosemium* and *Trifolium*, with the further subdivision of *Trifolium* into eight sections—*Glycyrrhizum* (2 species), *Paramesus* (2 species), *Lupinaster* (3 species), *Trifolium* (73 species), *Trichocephalum* (9 species), *Vesicastrum* (54 species), *Trifoliastrum* (20 species), and *Involucrarium* (72 species). In 2014, *Trifolium* phylogenetic analyses were conducted, based on highly unusual *Trifolium* plastomes [14].

The economic importance of this genus lies in its agricultural utilization. Historically, clovers, and especially red clovers, have been cultivated in rotation with other crops to maintain soil fertility due to their ability to establish a mutualistic relationship with root-nodulating and nitrogen-fixing bacteria. Their value was later diminished by the advent of nitrogen fertilizers, but the global need for sustainable and conservation agriculture is bringing this historical approach back into focus. Nowadays, many *Trifolium* species are extensively cultivated as fodder plants (*Trifolium pratense* L., *Trifolium repens* L., *Trifolium hybridum* L., and *Trifolium resupinatum* L.), and also as green manure crops to enhance soil fertility and sustainability [15]. Further knowledge about the genomes of both wild and cultivated clovers and an understanding of their evolution will prove to be of great benefit in the future of clover breeding. 

## 2. *Trifolium* Genomes, Chromosomes, and Chromosome Number Variation

Genetic information within the cells of plants and animals alike experience changes during evolution. Chromosome structure changes through rearrangements involving DNA fissions, fusions, duplications, deletions, insertions, inversions, translocations, and the expansion of repetitive sequences [16,17,18,19,20]. These rearrangements are further heightened by polyploidization events that multiply a single genome (autopolyploidy) or combine two or more divergent genomes (allopolyploidy). These events are accompanied and followed not only by an increase in chromosome sets, but also by dynamic and stochastic changes in genome organization, including altered gene expression or transposon activation as well as structural rearrangements [21,22,23]. Therefore, plant chromosomes and karyotypes show significant variability and even closely related species can differ greatly, and *Trifolium* is no exception. 

The *Trifolium* genus has small to medium genome sizes, ranging from 337.1 Mb in *Trifolium ligusticum* Balb. ex Loisel. to 5669.3 Mb in *Trifolium pannonicum* Jacq. per 1C value [24] and includes species with different chromosome numbers and ploidy levels. Chromosome numbers have been published for more than 184 *Trifolium* species [1,25,26,27,28] and the Chromosome Counts Database compiles 318 species for the *Trifolium* genus entry [29]. Agriculturally important *Trifolium* spp., their chromosome numbers, and their estimated genome sizes are summarized in Table 1. The ancestral chromosome constitution within the *Trifolium* genus is considered to be diploid, with 2n = 16 [3]. While 80% of the already analysed species have a basic chromosome number of x = 8, corresponding with the ancestral state, species forming aneuploid series (x = 7, 6, 5) have been identified in 31 species, 11 of which have both aneuploid and diploid or polyploid counts [3,30]. Ellison et al. [3] found an uneven distribution of aneuploidy across the phylogeny to be present in subgenus *Chronosemium*, sections of *Trichocephalum*, *Trifoliastrum*, and *Vesicastrum*, although most commonly in sections of *Trifolium* [3]. Vozárová et al. [31] found aneuploidy in sections of *Lupinaster* [31]. In sharp contrast to aneuploidy, polyploidy has been observed to be evenly distributed across the phylogeny, with all of the large clades having at least one inferred origin of polyploidy, the largest number (10) occurring in section *Involucrarium* [3]. In total, polyploidy has been observed in 24 species including tetraploidy, hexaploidy, octoploidy, dodecaploidy, and hexadecaploidy, and some species have been found to have both diploid and polyploid or have had different polyploid counts [3].

Perennial species in the *Trifolium* genus are known to be mostly self-incompatible. For instance, the agronomically important perennials *T. repens*, *T. pratense*, *T. hybridum*, and *Trifolium ambiguum* M. Bieb. have a gametophytic system of incompatibility determined by one locus, with both *T. pratense* and *T. repens* possessing large numbers of alleles [32]. The mating system of these self-incompatible crops leads to the development of highly heterozygous populations [33,34]. On the other hand, annual species, in general, are mostly self-compatible (e.g., *Trifolium subterraneum* L. and *Trifolium alexandrinum* L.), even though some may require a stimulating agent, such as insects or tripping, to produce high quantities of seeds [35,36].

The *Trifolium* chromosomes are small in size, usually metacentric or submetacentric [1,27,37,38,39,40]. To date, acrocentric chromosomes have been observed only in *Trifolium argentinense* Speg. [41]. Telocentric chromosomes have not yet been confirmed in the genus. These are generally rare in plants and eukaryotes, probably due to instability, which is attributed to the reduced centromere size and incompleteness of kinetochores that occur through the mis-division of centromeres in normal chromosomes.

## 3. Current State of Knowledge on *Trifolium* Genomes

Further research on clover genomes at the sequence level, through the localization of genes, was first enabled by genetic mapping. Since the beginning of the millennium, linkage maps have been published for three agronomically important cultivated clover species. A restriction fragment length polymorphism (RFLP) markers linkage map, high-density linkage maps with gene-associated microsatellite markers, and a consensus linkage map have been made available for red clover [42,43,44]. The latter has also been applied in white clover *T. repens* and *T. subterraneum* for comparative mapping [45,46,47].

In white clover, four independent genetic linkage maps were published [45,47,48,49], followed by an integrated linkage map published in 2013 [50]. The integrated map of *T. repens* includes 1109 loci, 18 candidate genes, and 1 morphological marker and covers 97% of the genome at a moderate density of one locus per 1.2 cM. The alignment of the integrated map with the *M. truncatula* genome assembly revealed substantial collinearity, and identified an interchromosomal rearrangement, whereby the *M. truncatula* chromosomes Mt-2 and Mt-6 were split across the *T. repens* chromosomes Tr-2 and Tr-6. In 2012, the first genetic linkage map for *T*. *subterraneum* was constructed showing great synteny between *T. subterraneum* and *T. pratense*, and proving transferability of *T. pratense* and *T. repens* molecular markers to *T. subterraneum* [46]. The advancement of next-generation sequencing assisted in acquiring sequences of numerous legume species, including clovers. To date, whole genome sequences of six clover species have been made available, and include both cultivated (*T*. *pratense*, *T*. *repens*) and wild (*Trifolium medium* L., *T*. *subterraneum*, *Trifolium occidentale* Coombe, *Trifolium pallescens* Schreb.) species.

The first clover draft genome sequence was made available for cultivated red clover (*T. pratense*, tetraploid variety Tatra) and was published by Ištvánek et al. [51]. The authors presented 314.6 Mbp of an estimated genome size 418 Mbp and annotated and predicted functions for 47,398 protein-coding genes, within a total of 64,761 predicted genes [30]. A year later, the genome sequence of another T. *pratense*, this time the diploid variety Milvus, was published [52]. Its assembly consists of 309 Mbp, with 164.2 Mb in seven chromosome-length sequences or pseudo-molecules. Among the annotated genes, 22,042 of a total 40,868 were anchored onto the seven chromosomes, making red clover the second forage legume with a genome assembly at a pseudo-molecule level just behind the model species *M*. *truncatula*. The authors also anchored the acquired data to the *M*. *truncatula* reference sequence [53] for the physical map construction and synteny analysis. Macrosynteny was observed in 248 synteny blocks carrying 12,278 gene pairs, with chromosomes 1 and 6 being almost entirely syntenic with *M*. *truncatula* chromosomes 1 and 7 and the remaining five chromosomes having large synteny blocks with two or three *M. truncatula* chromosomes [52]. The synteny between the red clover and *M*. *truncatula* is significantly less conserved than between the white clover and *M*. *truncatula*. The reason for this variability is most probably the difference in their basic chromosome numbers (white clover and *M*. *truncatula* x = 8, red clover x = 7) [47,52].

In 2014, a genome sequence of the wild zigzag clover *T*. *medium*, with an estimated genome size of 3154 Mbp [30], was published. To date, it remains the largest *Trifolium* genus sequencing project [54]. The authors assembled a partial genomic sequence of 492.7 Mb. The final assembly is very fragmented, due to, among other reasons, the large haploid size of the zigzag clover genome, its polyploid state, cross-pollination, and a high proportion of repetitive sequences equal to nearly half of the (46.67%) the genome size. Although the assembly was not sufficient for comprehensive annotation, the authors did provide a characterization and a comparative analysis of repeat content between *T. medium* and *T. pratense*.

Genome sequences of *T. subterraneum* were published in 2016 [55]. Those authors assembled genomic sequence of 471.8 Mb, covering 85.4% of the *T. subterraneum* genome and the annotated 30,543 of 42,706 predicted genes. A comparative analysis of *T. subterraneum* and of related legume species from the subfamily Papilionoideae revealed that the whole chromosome 1 in *T. subterraneum* is highly conserved across all analyzed species [55]. Moreover, the authors observed a clear syntenic relationship with *M. truncatula* chromosomes and revealed single synteny blocks for three pseudomolecules (Tsub-1 and Mt-1, Tsub-3 and Mt-3, and Tsub-5 and Mt-5). On the other hand, the authors observed the occurrence of duplication in *T. subterraneum* chromosome 2, compared to *M. truncatula* and *T. pratense* chromosome 2, suggesting that duplication occurs after the divergence of *T. pratense* and *T. subterraneum*. Alongside acquiring the genome sequences of *T*. *pratense* and *T*. *subterraneum*, the authors constructed high-density SNP linkage maps [52,55]

Genome sequences of an allotetraploid white clover were published in 2019 along with genome sequences of its extant relatives of the parental progenitors, namely, the western clover *T*. *occidentale* and pale clover *T*. *pallescens* [56]. Their assemblies spanned 437 Mb for western clover, 383 Mb for pale clover, and 841 Mb for white clover, accounting for 82%, 72%, and 72% of the estimated genome sizes (530, 534 and 1174 Mb), respectively. Based on the data, the authors estimate that *T. repens* originated ca 15,000 to 28,000 years ago during the last glaciation in Europe through multiple hybridization events in the contact zones of the progenitors in glacial refugia. Multiple allopolyploidization events indicate a high compatibility of the parental genomes and resulted in a noticeable diversity of newly developed *T. repens* populations. However, the authors show that subgenomes of the progenitors have maintained integrity and independence as well as gene expression activity. Overall, the allopolyploidization event of two highly specialized species enabled the newly developed species *T. repens* to colonize different niches and expand worldwide. 

## 4. Hybridization in the Clover Genus

Ellison et al. [3] carried out a comprehensive DNA-based phylogenetic analysis, and only five or six instances of apparent hybrid speciation were found [3]. Overall, interspecific hybridization does not appear to play a significant role in the evolution of clovers, and this may well be associated with their predominant adaptation to insect pollination [57].

Despite the strong genetic barriers to interspecific hybridization, origin through hybridization was suggested in the cases of *Trifolium dubium* Sibth., *T*. *repens*, *T*. *pannonicum*, and *T*. *medium* [3,54]. To date, parental origin was identified in tetraploids white clover (*T*. *repens*), derived from hybridization of *T*. *occidentale* and *T*. *pallescens* [3,34], and suckling clover *T*. *dubium*, combining the genomes of *Trifolium campestre* Schreb. and *Trifolium micranthum* Viv. [58]. The ancestry of *T*. *pannonicum* seems to be complex, given that Ellison et al. [3] suggested its parental species to be distantly related to section *Trifolium*—one of which is probably related to *Trifolium patulum* Tausch. and *Trifolium squamosum* L., with the other unknown or perhaps extinct. For *T*. *medium*, its allopolyploid origin was proposed based on the existence of different types of centromeric repeats [54]. On the other hand, legumes reveals an exceptional diversity and promiscuity of satellite repeats in general [59,60,61].

Economically important clovers, such as white clover (*T*. *repens*) and red clover (*T*. *pratense*) are subject to breeding programs to enhance traits such as yield, flowering, nutrient use efficiency, stress tolerance, and resistance to specific diseases. The introduction of any specific trait via artificial interspecific hybridization is difficult to accomplish, however, and has generally only been achieved between closely related taxa, often with the aid of embryo-rescue techniques or the use of bridge species [62,63], as previously reviewed by Abberton [33]. Despite the great efforts directed to the introgression of desirable traits into cultivated clovers by crossing, only a few specific attempts have been successful.

To date, 11 related clover species were discovered to be capable of being integrated into the wider gene pool of white clover by interspecific hybridization (*Trifolium nigrescens* Viv., *T*. *occidentale*, *Trifolium isthmocarpum* Brot., *Trifolium uniflorum* L., *T. ambiguum*, *T*. *hybridum*, *Trifolium thalii* Vill., *T*. *pallescens*, *Trifolium montanum* L., *Trifolium argutum* Banks & Sol., and *Trifolium semipilosum* Fresen.) [63]. In the case of red clover, considerably less work has been carried out in its artificial hybridization with other taxa. Red clover has been successfully hybridized with five species so far (*Trifolium sarosiense* Hazsl., *Trifolium alpestre* L., *T*. *ambiguum*, *Trifolium diffusum* Ehrh., and *T*. *medium*; reviewed by Abberton [33]), but the only viable progeny has resulted from hybrids between tetraploid *T*. *pratense* cv. Tatra and octoploid *T*. *medium* [64,65]. As the ability to produce interspecific hybrids reflects the compatibility of parental genomes, further knowledge about genomes and of their similarities and differences can strongly support breeding efforts in *Trifolium*.

## 5. Chromosome Identification in *Trifolium*

In legumes with large chromosomes, such as *Pisum sativum* L. or *Vicia faba* L., individual chromosomes can be distinguished by ordinary karyotyping or banding methods [66,67], although the process is rather complicated in species with small chromosomes, such as *Trifolium*. Both repetitive and low- or single-copy sequences are important tools for chromosome identification in cytogenetic studies. Usually, a mix of different probes is used, which can include localizing ribosomal DNA (rDNA) sites, telomeric probes, large plasmid, bacteriophage, or bacterial artificial chromosomes (BACs) containing specific single-copy or repetitive inserts. Based on 5S rDNA, 25S rDNA, and seven bacterial artificial chromosome probes containing microsatellite markers with a known position, a cytogenetic map has been constructed for red clover (Figure 1).

The easy design and production of oligonucleotide libraries has presented new opportunities to plant cytogenetics. Recently, an oligonucleotide barcode system was developed to identify all cowpea and common bean chromosomes [69,70]. Despite the availability of genome sequences for selected *Trifolium* spp., however, oligonucleotide libraries have not yet been exploited for *Trifolium* research.

## 6. Chromosomal Distribution of Ribosomal DNA Genes

Because ribosomal genes are among the best-researched regions of eukaryotic genomes, fluorescence in situ hybridization (FISH) analyses, using rDNA genes as probes, have been conducted in numerous plant species, including *Trifolium*. The 35S and 5S ribosomal genes are located independently in one or several loci as tandem repeats, ranging from hundreds to thousands of copies in higher vascular plant genomes [71]. While polycistronic gene 35S consists of 18S-5.8S-25S rDNA and occurs on chromosomal regions known as nucleolus organizer regions (NORs), the 5S rDNA gene is usually independent from NORs [72,73]. Ribosomal genes have undergone rapid evolution in their means of altering the number of copies and their localization on chromosomes [74,75,76,77]. Therefore, rDNA genes have been proven to act as excellent cytogenetic markers for karyotype analysis, and they have been widely used to examine and understand phylogenetic relationships, chromosomal organization, and evolution in many plant species. 

Roa and Guerra [78,79] found that, in angiosperms rDNA sites, most often number one or two 45S and one 5S per haploid genome. The localization of 45S rDNA sites was observed preferentially on the short arm and in the terminal region of chromosomes in general, but genera with predominant proximal localization were found in some families, including Fabaceae (*Arachis*, *Lens*). On the other hand, 5S rDNA localization varies in different angiosperm families. In Fabaceae, 5S rDNA is preferentially found in the proximal region.

To date, the numbers and positions of rDNA loci on chromosomes have been reported for 42 *Trifolium* species (Table 2; adapted from Vozárová et al. [31]). Based on ancestral state reconstruction, Vozárová et al. [31] suggested the occurrence of one 5S and one 26S locus per haploid genome separately as an ancestral condition for the whole genus. The ancestral karyotype referencing the basic chromosome number and rDNA loci constitution may resemble the karyotype of *T. diffusum* (Figure 2).

## 7. Other Repetitive and Single-Copy Markers

The repetitive elements of both a satellite and dispersed character have been used as cytogenetic markers to define karyotypes in plants because these sequences constitute large proportions of most plant genomes. Even though in some species the satellite repeats can represent as much as 10–20% of the genome, the bulk of the repeat fraction usually consists of mobile elements, specifically long terminal repeats (LTR) retrotransposons [16,83,84,85,86,87,88]. The accumulation of LTR retrotransposons along with multiple rounds of polyploidization events are crucial in genome size expansion [18,89]. Repetitive elements are often clustered in heterochromatic regions, such as pericentromeres, and are crucial for maintaining genome stability. Pericentromeric regions and their repetitive content have been researched in *M*. *truncatula* and *Glycine max* (L.) Merill. model legumes related to *Trifolium* [90,91].

Most satellite repeats occupy centromeric, pericentromeric, subtelomeric, or telomeric regions. Typical plant centromeres are composed of centromere-specific satellite repeats, and often act as the dominant component, as well as retrotransposons [92]. In the 1990s, the C-banding technique was used to examine highly repetitive sequences in plant chromosomes. C-banding discriminates the constitutive heterochromatin, which is generally found in the centromeric, although sometimes in the telomeric or interstitial, region and consists largely of highly repetitive DNA. Zhu et al. [93] reconstructed a *T. repens* karyotype using C-banding for constitutive heterochromatin identification and by analyzing a 350 bp tandemly repeated DNA sequence. The analyses revealed 13 pairs of metacentric and 3 pairs of submetacentric chromosomes in *T. repens* karyotype. The C-bands were identified around the centromeric regions of 8 pairs of chromosomes, and no terminal or interstitial C-bands were observed. A probe derived from 350 bp tandemly repeated DNA sequence, hybridized to a centromeric region of 12 chromosome pairs, representing some of the chromosomes that correspond with C-bands. Similarly, karyotypes with C-bands were constructed by Bucknell [94] in *Trifolium hirtum* All. and *T. incarnatum.* The author observed C-bands in the centromeric and pericentromeric regions of *T. hirtum* and in the centromeric region and proximal-end of the satellite body in *T. incarnatum*. 

Ansari et al. [95] analyzed the structure of a 350 bp centromeric satellite repeat identified by Zhu et al. [93], and its presence in the genome of 17 *Trifolium* species and subspecies, revealing a lineage-specific centromeric satellite repeat unique to species from section *Lotoidea* with Mediterranean origins. An *in situ* hybridization probe, derived from the analyzed centromeric satellite repeat, hybridized to all chromosomes of *T. uniflorum*, *T. occidentale*, and *Trifolium nigrescens* ssp. *petrisavii* (Clem.) Holmboe and ssp. *nigrescens*, to four chromosome pairs in *T. pallescens* (note that together with *T. occidentale* it corresponds to the hybridization pattern in *T. repens* [93]), to two chromosome pairs in *T. isthmocarpum*, and to one chromosome pair in *Trifolium michelianum* Savi, *T. montanum*, and *T. ambiguum*. However, C-banding conducted by those authors revealed the presence of C-bands on all chromosomes of *T. ambiguum*, thus indicating the coexistence of more than one family of the centromeric satellite repeats in this species. Similarly, the coexistence of two or more centromeric satellite repeats can be expected in other species whereby a probe derived from the analyzed centromeric satellite repeat hybridizes to only some chromosome pairs. According to the so-called “library hypothesis” proposed by Fry and Salser [96], related species share a collection of satellite sequences that may expand and contract in evolution. Ansari et al. [95] suggested that at least a partial homogenization and the amplification of this repeat family occurred before the radiation of this lineage and, during its evolution, diverse expansion and the reduction of specific satellite repeats resulted in differentiation and the presumable coexistence of more than one family of centromeric satellite repeats in some species.

The availability of genome sequences enables the further characterization of repetitive elements in plant genomes. Dluhošová et al. [54] conducted a thorough comparison between red clover and zigzag clover sequencing reads [54]. Repeat content characterization revealed a striking difference between the overall repeat content in both species and in the prevalence of individual DNA transposon lineages. The proportion of fully annotated repetitive elements was 33% in *T. pratense* and almost 47% in *T. medium*. However, due to a possible underestimation of the overall repeat content caused by the low number of reads included in the analysis, the repetitive elements may comprise as much as 70% of the *T. medium* genome. The major differences in the prevalence of individual repetitive elements showed a reduction of Ty1/Copia (12.22% in *T. pratense* and 7.80% in *T. medium*) and DNA transposons (6.07% in *T. pratense* and 2.89% in *T. medium*) and a remarkable rate of expansion (6.65% in *T. pratense* and 26.29% in *T. medium*) of Ty3/Gypsy retrotransposons in the genome of *T. medium*, compared to *T. pratense*. The authors also identified 7 and 45 species-specific clusters for *T. pratense* and *T. medium*, respectively, of which 6 and 11 were successfully validated. The newly discovered *Trifolium*-specific tandem repeats contained one centromeric and three pericentromeric repeats in *T. pratense* and two centromeric, one pericentromeric, and one subtelomeric repeat in *T. medium*. Based on the hybridization pattern of probes derived from 5S and 26S rDNA and on all the newly discovered *T. medium*-specific tandem repeats, the authors divided 64 *T. medium* chromosomes into 11 categories (Figure 3). Moreover, based on the presence of the two identified centromeric repeats on only half of the chromosomes, an allopolyploid origin of *T*. *medium* was suggested. A considerable variability and promiscuity in the satellite repeats has been observed in many Fabeae species generally [59,60,61], however, and the coexistence of more than one family of centromeric satellite repeats, emerging from the diverse expansion and reduction of specific satellite repeats in evolution, has already been proposed for some *Trifolium* species (e.g., *T. ambiguum* and *T. pallescens*) [95]. In addition to *T. pratense*, *T. medium*, and *T. repens*, centromere-specific satellite repeats have been found in many related legumes, such as cowpea, soybean, chickpea, common bean, or alfalfa and their related species [97,98,99,100,101,102,103].

In large and complex plant genomes, a remarkable proportion of which can consist of repetitive sequences, unique motifs suitable as cytogenetic markers are restricted to regions that are only a few kb long, and that usually coincide with genes [104,105,106]. Successful reports of detecting single-copy gene sequences in legumes related to *Trifolium* include a legumin gene that us 13.5 kb long in pea [107], and a β-tubulin gene sequence that is 10 kb long in alfalfa [108]. The localization of single-copy DNA sequences has been hampered, however, by technical difficulties, mostly relating to the low sensitivity of short-length probes [109]. Since the turn of the century, this challenge has been overcome by advances in single-gene FISH protocols, modern microscopy resolution and by using large-insert DNA clones, usually BACs, anchored by the targeted small DNA probes [110,111,112,113,114,115,116].

Since then, BAC clones have been used for the physical mapping of single-copy sequences in *T*. *pratense*, including in constructing *T. pratense* cytogenetic maps [43,68] and in related legumes, including the model species *M*. *truncatula* [90] and *L*. *japonicus* [117] as well as the agronomically important species *Phaseolus vulgaris* L. [118,119] and *Glycine* spp. [99].

## 8. Comparative Mapping in Legumes

The hybridization of species-specific elements to their related species might provide insight into a species’ evolution and reveal rearrangements. The hybridization of probes to large chromosome regions is convenient for these analyses. Despite the common utilization of large chromosome segments or whole chromosomes acquired by flow sorting or microdissection in animal and human cytogenetics [120], these techniques are not applicable in plants due to their high complexity and the large number of repeated structures of various types. Instead, cytogenetic analyses in plants have been developed to exploit synthetic oligonucleotide libraries (oligo-FISH, oligo-painting) and high-capacity DNA vectors, such as BACs, carrying large inserts of DNA (BAC-FISH, BAC-painting).

As mentioned above, BAC-FISH is a powerful tool for chromosome identification and has been successfully used to mark specific chromosomes in many plant families, including legumes. The application of BACs has been extended to identify large chromosome regions, such as chromosome arms or whole chromosomes (BAC-painting). However, the availability of whole genome sequences, obtained from BAC to BAC sequencing and a sufficient pool of low- or single-copy BAC clones, is required for the successful development of chromosome BAC probes. Thus, BAC-painting is suitable for species with small genomes and low proportions of repetitive fraction, and to date it remains established only in the family Brassicaceae [121,122] and for *Brachypodium* in the family Poaceae [123], even though attempts have bene made to extend this method to other families [124,125].

Significant progress has recently been made in karyotype research in plants for the application of artificially synthetized oligonucleotides (oligo-FISH, further extended to oligo-painting) [126]. Short and unique oligonucleotides (usually 45–50 bp) can be computationally identified and designed specifically for a chromosomal region or even an entire chromosome. The hybridization of specific oligo pools to their related species has proven to be a powerful tool to reveal karyotypic and chromosomal evolution [126,127,128,129,130,131,132,133]. To date, this method has been used in a single legume species, *Arachis* [134], but the availability of the genome sequences of selected *Trifolium* spp. Creates opportunities for utilizing oligonucleotide libraries in *Trifolium* karyotype and evolutionary research.

## 9. Conclusions and Future Prospects

Improving clover productivity as a means of boosting yields and nitrogen fixation efficiency is therefore a central focus of plant breeders today. Information is mainly limited to the related legume model species, however, and unraveling the genome organization and understanding the evolution of clover are essential for greater breeding efficiency. 

Advances in clover research within the genomics era have assisted in the development of an impressive array of genomic resources, including complete genome sequences of some clovers and related legumes. As high-throughput sequencing has revolutionized genome sequencing with its ultralow cost and overwhelmingly large data output, more and more new plant species sequences, as well as species’ resequences, supported by a large range of bioinformatic tools, provide us with more data applicable for more efficient breeding strategies. The combination of genomic and bioinformatic data with molecular cytogenetics may provide a more developed understanding of plant genomes in general.

Ribosomal DNA and other repetitive sequences have been widely used as plant cytogenetic markers, and recently, the development of large-DNA clones carrying target sequences, such as BACs, has facilitated the easier localization of low- or single-copy DNA sequences [110,111,112,113,114,115,116]. BAC-FISH has been applied successfully in clover karyotype characterization, and cross-species BAC-FISH has helped to identify chromosome structure and rearrangements in clover relatives such as the common and lima bean [135]. However, the extension of BAC-FISH to the BAC-painting of large chromosome regions is suitable only for species with small genomes and low proportions of repetitive fractions, and it has not been successfully established beyond crucifers [121,122], with the singular exception of Brachypodium [123].

The remarkable progress in plant genome research relating to reference sequences production and artificial DNA synthesis has provided an alternative chromosome painting technique. In silico designed and artificially synthesized oligonucleotide pools have already been applied successfully in various plant species to characterize chromosomal rearrangements [126,127,128,129,130,131,132,133]. The availability of the Trifolium genome and reference sequences means that the adoption of oligo painting within this genus, and the legume family more generally, is both possible and to be expected. 

## Figures and Tables

**Figure 1 plants-10-02518-f001:**
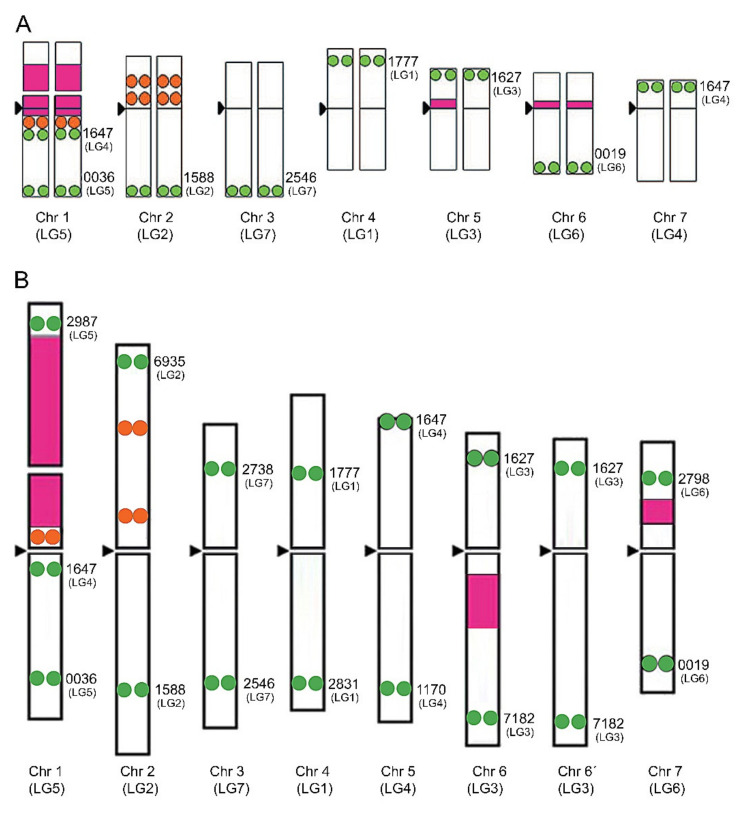
Cytogenetic map of *T. pratense* based on the hybridization pattern of probes derived from 5S rDNA (orange circles) and 26S rDNA (pink boxes) and localization of 7 (**A**) or 14 (**B**) BAC clones corresponding to chromosome-specific microsatellite markers (green circles) (adapted from Sato et al. [43] and Kataoka et al. [68].

**Figure 2 plants-10-02518-f002:**
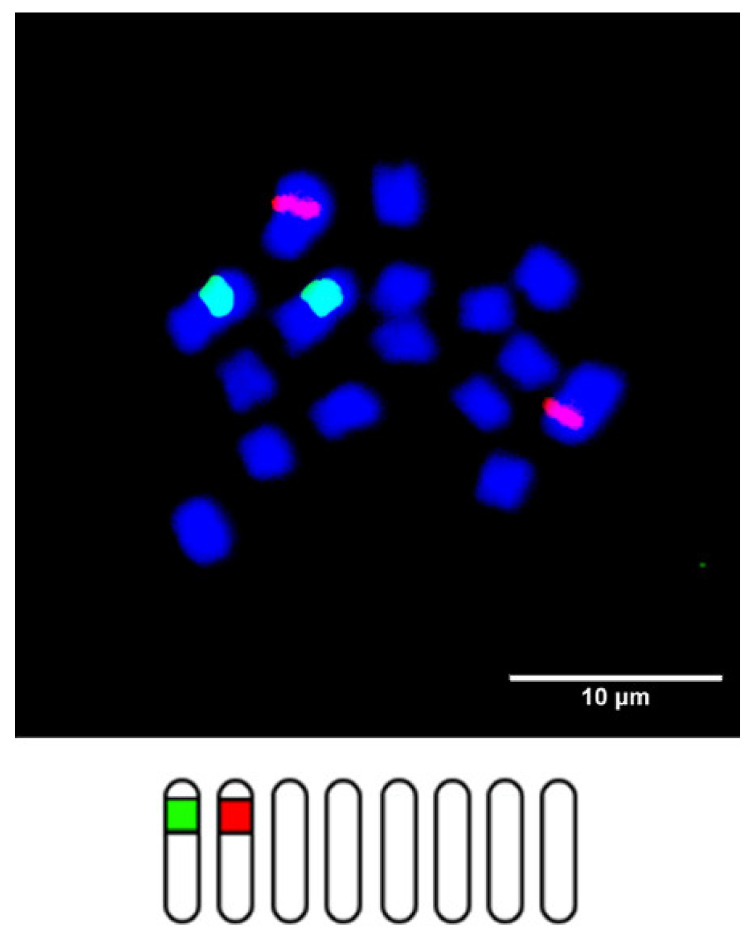
Fluorescence in situ hybridization image and schematic karyotype of *T. diffusum* with 5S (red) and 26S (green) rDNA probes suggested to represent the ancestral state in the *Trifolium* genus (adapted from Vozárová et al. [31]).

**Figure 3 plants-10-02518-f003:**
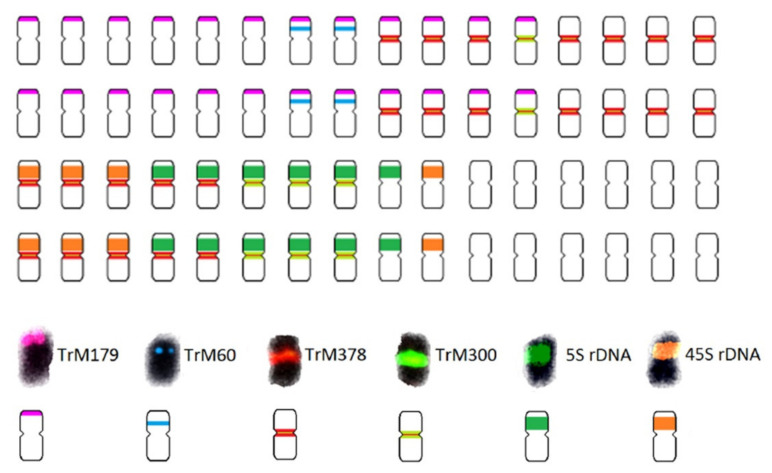
Schematic karyotype of *T. medium* based on the hybridization pattern of probes derived from 5S (dark green) and 26S (orange) rDNA, subtelomeric repeat TrM179 (pink), pericentromeric repeat TrM60 (blue), and two centromeric repeats TrM378 (red) and TrM300 (light green). Centromeric repeats Trm378 and TrM300 are colocalized on the same chromosomes, 24 chromosomes have a higher proportion of TrM378 while 8 chromosomes have a higher proportion of TrM300 (adapted from Dluhošová et al. [54]).

**Table 1 plants-10-02518-t001:** Selected important *Trifolium* crops, their origins, chromosome numbers, and genome sizes.

Scheme 2.	Common Name	Section	Origin	2n ^1^	1C Content (Mb) ^2^
*Rifolium resupinatum* L.	Persian clover	Vesicastrum	Eurasian	2x = 14, 16, 4x = 32	507.6
*Trifolium incarnatum* L.	Crimson clover	Trifolium	Eurasian	2x = 14	652.3
*Trifolium repens* L.	White clover	Trifoliastrum	Eurasian	4x = 32	546.7
*Trifolium pratense* L.	Red clover	Trifolium	Eurasian	2x = 14, 4x = 28	417.6
*Trifolium alexandrinum* L.	Egyptian clover	Trifolium	Eurasian	2x = 16	541.8
*Trifolium subterraneum* L.	Subclover	Trichocephalum	Eurasian	2x = 16	542.8
*Trifolium tembense* Fres.	Tembien clover	Vesicastrum	African	2x = 16	818.6

^1^ Ellison et al. [3]. ^2^ Vižintin et al. [30]

**Table 2 plants-10-02518-t002:** Reported chromosome numbers 5S and 25S rRNA loci numbers in *Trifolium* species (adapted from Vozárová et al. [31]).

Subgenus/Section	*Trifolium* Species	2n	Loci Number per 2n		Reported in
			5S	25S	
**CHRONOSEMIUM**					
	*T. aureum*	2x = 16	4	2	Vozárová et al. [31]
		2x = 14	4	2	
	*T. badium*	2x = 14	2	4	
			2	2	
	*T. campestre*	2x = 14	2	2	Ansari et al. [58]
	*T. micranthum*	2x = 16	2	2	
	*T. dubium*	4x = 30	4	4	
**TRIFOLIUM**					
TRIFOLIUM	*T. alpestre*	2x = 16	10	2	Vozárová et al. [31]
			11	2	
	*T. arvense*	2x = 14	2	2	
	*T. bocconei*	2x = 12	2	2	
	*T. cherleri*	2x = 10	4	10	
	*T. diffusum*	2x = 16	2	2	
	*T. hirtum*	2x = 10	6	2	
	*T. ligusticum*	2x = 12	2	2	
		2x = 14	2	2	
	*T. pallidum*	2x = 16	4	2	
	*T. purpureum*	2x = 14	2	2	
	*T. rubens*	2x = 16	4	2	
	*T. squamosum*	2x = 16	4	2	
	*T. stellatum*	2x = 12	4	2	
		2x = 14	4 (2w)	2	
	*T. pannonicum*	16x = 128	16	16	
	*T. pratense*	4x = 28	8	8	Dluhošová et al. [80]
		2x = 14	4	5	Sato et al. [43]
	*T. medium*	8x = 64	12	8	Dluhošová et al. [80]
TRICHOCEPHALUM	*T. subterraneum* subsp. *subterraneum*	2x = 16	2	4 (2w)	Vozárová et al. [31]
	*T. subterraneum*subsp. *subterraneum*	2x = 16	2	2	Falistocco et al. [81]
	*T. subterraneum* subsp. *brachycalycinum*	2x = 16	2	4 (2w)	
	*T. israeliticum*	2x = 12	10	4	
VESICASTRUM	*T. fragiferum*	2x = 16	2	2	Vozárová et al. [31]
	*T. resupinatum*	2x = 16	2	2	
		2x = 14	2	2	
	*T. spumosum*	2x = 16	2	2	
			4	2	
TRIFOLIASTRUM	*T. glomeratum*	2x = 16	2	2	Vozárová et al. [31]
	*T. montanum*	2x = 16	2	2	
	*T. occidentale*	2x = 16	4	2	Ansari [82]
	*T. pallescens*	2x = 16	2	2	Vozárová et al. [31]
	*T. thalii*	2x = 16	2	2	
	*T. repens*	4x = 32	4	2	Ansari [82]
	*T. uniflorum*	4x = 32	4	4	
	*T. nigrescens* subsp. *nigrescens*	2x = 16	2	2	
	*T. nigrescens* subsp. *petrisavii*	2x = 16	2	2	
	*T. ambiguum*	2x = 16	2	2	
	*T. hybridum*	2x = 16	2	2	
	*T. isthmocarpum*	2x = 16	2	6	
INVOLUCRARIUM	*T. chilense*	2x = 16	4	2	Vozárová et al. [31]
	*T. microdon*	2x = 16	2	2	
	*T. microcephalum*	2x = 16	16	16	
			16	2	
PARAMESUS	*T. glanduliferum*	2x = 16	4	2	Vozárová et al. [31]
			5 (1w)	2	
			4	2	
	*T. strictum*	2x= 14	2	2	
LUPINASTER	*T. lupinaster*	4x = 28	8	4	Vozárová et al. [31]
		4x = 32	8	4	

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
