# Peer review of "Chromosome and Genome Diversity in the Genus Trifolium (Fabaceae)"

_plants, 2021, doi:10.3390/plants10112518_

Round 1

Reviewer 1 Report

This is a nice, comprehensive review on the current understanding of Trifolium genomes and chromosomes, as well as some associated methodologies. I generally enjoy reading it.

Just one minor revision from me is that on Line 128 and 132 (and maybe other similar context too), please add the word "genome" or "whole genome" or "draft genome" before the word "sequence". The sequence of Trifolium had been published long before the genome.

Author Response

Thank You for Your time reviewing our manuscript. Y

ou are absolutely right and we have revisioned the whole text so it is clear now where we are addressing whole genome sequences. 

Kind regards,

Eliska Lukjanova

Reviewer 2 Report

The manuscript "Chromosome and genome diversity in the genus Trifolium (Fabaceae)" summarizes the current state of knowledge on Trifolium genomes and chromosomes. I found the review well designed and readable in spite of some language errors. The illustration and tables give a good insight into the topic of the review. However, minor edits are required for this manuscript as follows:

  1. The English language in the manuscript needs revision. the authors used synonymous words at many places as "Molecular cytogenetics and cytogenomics" and "Nowadays, therefore, .......".
  2. The authors declared in the abstract that they are review methodologies that have contributed to Trifolium research, I see that point is not covered concisely as they did not describe methods used for the diversity analysis as the morphological, cytological, biochemical markers but they mentioned some studies that done using some molecular markers. So please edit the description of the aim of the manuscript in the abstract.
  3. Please add the following citation in line 363 "Ebeed, H. T. (2019). Omics approaches for developing abiotic stress tolerance in wheat. In Wheat Production in Changing Environments (pp. 443-463). Springer, Singapore."

Author Response

Firstly, thank You for Your time spent revising our manuscript.

We appreciate Your comments. To the points given:

  1. Thank You for this suggestion, we revised the manuscript in order to get rid of language errors and redundant terms.
  2. Thank You, we can see Your point and we agree and edited the abstract. 
  3. We added this citation to the manuscript.

Kind regards,

Eliska Lukjanova

Reviewer 3 Report

This is a well written and structured review article that fully covers the knowledge about cytology and cytogenetics of the genus Trifolium and especially of the most important crops. I therefore recommend acceptance of this manuscript for publication in Plants.

Author Response

Thank You very much for Your time spent reviewing our manuscript and for Your kind words.

Kind regards,

Eliska Lukjanova

Reviewer 4 Report

In this work, the authors realized an intensive review on Trifolium molecular cytogenetics and cytogenomics to produce an interesting manuscript. The text is well written, supported by sufficient data and references and interpreted the easy way to the reader. The weakest aspect of the paper in object probably relates with the using of verbose paragraphs, especially in the conclusion section. There are other important points to be addressed, which I have tried to focus to in the attached PDF.

Author Response

Firstly, thank You for Your time spent revising our manuscript and especially for the comments included in the attached PDF.

We agree with comments and suggestions made and tried to revise our manuscript as You suggested e.g. ommited redundant sentences in the abstract and conclusion, adjusted vocabulary used replacing cultured to cultivated. We also omitted citations in the text reference to figures and left only the citations in the legends as You pointed out and adjusted the confusing and innacurate formulations in chapters 5 and 7. 

Kind regards,

Eliska Lukjanova